# Learning Semantic-aware Normalization for Generative Adversarial Networks

**Heliang Zheng[1]\*, Jianlong Fu[2], Yanhong Zeng[3]\*, Jiebo Luo[4], Zheng-Jun Zha[1]†**
[1]University of Science and Technology of China, Hefei, China
[2]Microsoft Research, Beijing, China
[3]Sun Yat-sen University, Guangzhou, China
[4]University of Rochester, Rochester, NY
[1]zhenghl@mail.ustc.edu.cn, [2]jianf@microsoft.com,[3]zengyh7@mail2.sysu.edu.cn
[4]jluo@cs.rochester.edu, [1]zhazj@ustc.edu.cn

## Abstract

The recent advances in image generation have been achieved by style-based image generators. Such approaches learn to disentangle latent factors in different image scales and encode latent factors as "style" to control image synthesis. However, existing approaches cannot further disentangle fine-grained semantics from each other, which are often conveyed from feature channels. In this paper, we propose a novel image synthesis approach by learning **S**emantic-**a**ware **r**elative **i**mportance for feature channels in Generative Adversarial Networks (SariGAN). Such a model disentangles latent factors according to the semantic of feature channels by channel-/group- wise fusion of latent codes and feature channels. Particularly, we learn to cluster feature channels by semantics and propose an adaptive group-wise Normalization (AdaGN) to independently control the styles of different channel groups. For example, we can adjust the statistics of channel groups for a human face to control the open and close of the mouth, while keeping other facial features unchanged. We propose to use adversarial training, a channel grouping loss, and a mutual information loss for joint optimization, which not only enables high-fidelity image synthesis but leads to superior interpretable properties. Extensive experiments show that our approach outperforms the SOTA style-based approaches in both unconditional image generation and conditional image inpainting tasks.

## 1 Introduction

Image generation has achieved significant progress in recent years as generative adversarial networks (GAN) [1] attract a lot of attention and develop rapidly [2, 3, 4, 5, 6, 7, 8, 9, 10]. Notably, the recent success of GANs takes advantage of progressive growing for the generator. Such architectures make it possible to learn disentangled image styles from different spatial resolutions in the generator [2, 4, 11, 12], e.g., styles for global structures in low-resolution layers and styles for local details in high-resolution layers. Particularly, StyleGAN embeds latent codes into different spatial resolutions to control the scale-aware image styles during synthesizing [2, 12]. For example, StyleGAN is able to change the color schema of a generated human face by adjusting the style codes in high-resolution layers, while keeping the style of the global structure synthesized in low-resolution layers unchanged. Through such a scale-aware disentangled representation learning, StyleGAN yields state-of-the-art results for high-resolution image generation.

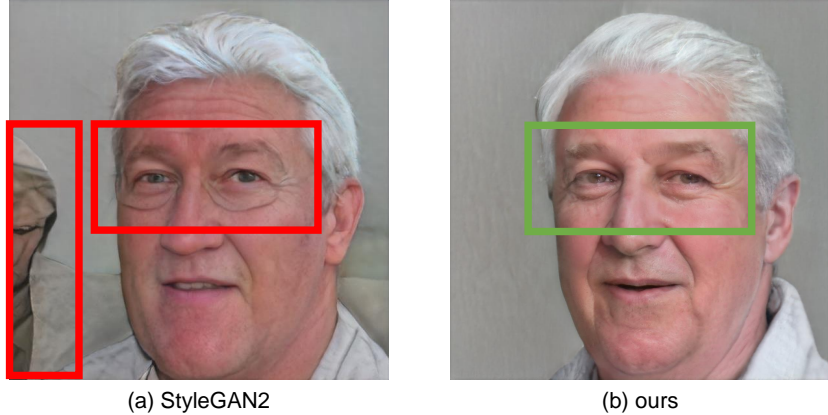

(a) StyleGAN2                    (b) ours

Figure 1: An example to show that semantic disentangling helps to generate more realistic images. For the image generated by (a) StyleGAN2, artifacts can be observed around the eyes, and the background is also unnatural. (b) Our model can remove such artifacts by learning to disentangle semantics (e.g., eyes and glasses).

However, the disentanglement in StyleGAN is limited, which makes it hard to further disentangle fine-grained semantics. Specifically, StyleGAN tends to change the image styles controlled by layers with different spatial resolutions, while it is difficult to change a specific style in the same layer. Inspired by the recent study on interpretable learning that different feature channels are closely related to different semantics [13, 14] and feature channels with the same semantics can be grouped together by their similarity [15, 16], we propose to further disentangle fine-grained semantics from each other by leveraging the semantics of feature channels. Specifically, after grouping feature channels into different groups by semantics, we propose to control the styles of different semantics independently.

To enable such a fine-grained semantic disentanglement, we propose to learn semantic-aware relative importance for channel groups in GAN (SariGAN) from deep image features. First, we design a similarity-based grouping module (SGM) to cluster channels with the same semantics together according to their similarity. The SGM separates different semantics from each other, which enables independent control for different semantic groups in subsequent style embedding operations. Second, we embed the input latent code into semantic-aware intermediate latent space (i.e., intra-group and inter-group) by learning a mapping network. Specifically, the intra-group codes modify the relative importance of features within each group for the subsequent convolution operation via an AdaIN operation. Meanwhile, the inter-group codes are used to control the relative importance of different groups by the proposed adaptive group-wise Normalization (AdaGN). Such channel-/group- wise fusions integrate the semantic group information into latent space and enables semantic disentangling for latent factors. Finally, the full model is jointly optimized by a non-saturating logistic GAN loss, a channel grouping loss and a mutual information loss [17]. Through such a design, the proposed SariGAN is able to disentangle fine-grained semantics and control the styles of specific semantics during generation.

In summary, our main contribution is to propose learning semantic-aware relative importance for fine-grained semantic disentanglement in GAN. We conduct both quantitative comparisons and qualitative analysis on several unconditional image generation benchmarks. Experimental results show that SariGAN can not only realize high-fidelity image synthesis but also has superior interpretability. Moreover, extensive experiments on conditional image inpainting also show improvements in generalization for SariGAN.

## 2 Related Work

### 2.1 Style-Based Generator

Since the style-based generator (i.e., StyleGAN [2]) was proposed by Karras et al., it has set a new state-of-the-art performance for unconditional image generation task and attracted a lot of attention [12, 16, 18, 19]. For example, Yuri et al. propose a way to distill a particular image manipulation

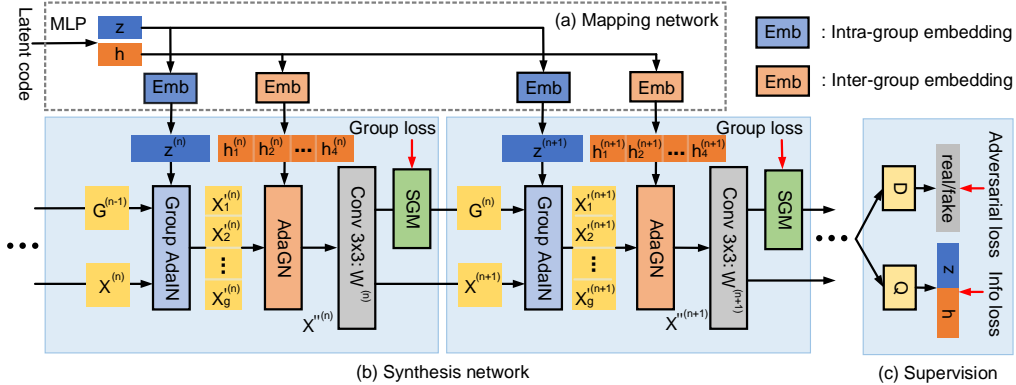

Figure 2: An overview of the proposed SariGAN. (a) Mapping network, which embeds the latent code to semantic-aware intermediate latent space (i.e., intra-group $\mathbf{z}$ and inter-group $\mathbf{h}$). (b) Synthesis network, which can achieve semantic-aware control by embedding $\mathbf{z}$ and $\mathbf{h}$ via Group AdaIN and AdaGN, respectively. (c) Supervision network, which consists of a stream $D$ to conduct adversarial loss by predicting real or fake, and a stream $Q$ to conducts a mutual information loss (info loss) by reconstructing $[\mathbf{z}, \mathbf{h}]$ [17].

of StyleGAN into an image-to-image network [19]. And Edo et al. propose a simple and effective method for making local edits to a target output image based on StyleGAN [16]. In general, style-based generators rely on progressive growing by initially focusing on low-resolution layers and then slowly shifting focus to finer details on high-resolution layers as the training proceeds [4]. Based on the progress growing, style-based generators are able to first output low-resolution images that are not affected significantly by high-resolution layers. Style-based generators take advantage of this property and propose to disentangle latent codes into "styles" by a mapping network to control each layer of the generator via AdaIN [20]. In AdaIN, layer activations are first normalized to zero mean and unit deviation and are further denormalized by modulating the activation inferred from the "styles". Through such a design, style-based generators are able to disentangle the latent factors for scale-aware control in image generation.

Karras et al. claim that the disentangled representation learning in StyleGAN helps in better image synthesis. There are increasing numbers of works on improving disentanglement for generative models [17, 21, 22, 23]. InfoGAN proposes to disentangle latent codes via maximizing the mutual information between the latent variables and the observation [17]. Despite promising results, InfoGAN is limited to low-resolution and simple images (e.g., digits). Other researchers propose to disentangle latent codes by using additional supervision (e.g., labels [19, 24], 3D prior [25, 26]). In this paper, we propose to take advantage of channel semantics to disentangle fine-grained semantics in a self-supervised manner, which leads to high-fidelity image synthesis and superior interpretability.

## 2.2 Interpretable Representation for Channels

Interpretable learning for generative models is an important topic that studies the reasonableness and reliability of image generation [14, 13, 27, 28, 29]. Some works proposed to conduct and analyze basic visual transformations by navigating in GAN latent space [28, 29]. Others proposed to visualize and understand GANs by modifying relative convolutional channels [14, 16]. Bau et al. found that channels in GAN are interpretable, which are closely related to object semantics [14]. Leveraging the semantic attributes of channels, Bau et al. are able to interactively manipulate objects in a scene by editing corresponding channels. And Collins et al. take one step further to cluster channels of a well-trained generator to achieve semantic-specific image editing [16]. In this paper, we make use of the interpretability of channels and propose a similarity-based grouping module (SGM), which can conduct channel grouping effectively and efficiently in generation models.

## 3 Semantic-aware Disentanglement

In this section, we introduce the proposed Semantic-aware Relative Importance GAN (SariGAN). First, we conduct similarity-based grouping in each layer of the generator to separate different semantics from each other. Thus we can achieve independent control for different semantic groups

in subsequent style embedding operations. Specifically, we learn a mapping network to embed latent code into semantic-aware intermediate latent space (i.e., intra-group and inter-group). Such embedded latent codes control intra-group and inter-group relative importance for feature channels via a group adaptive instance normalization (Group AdaIN) and a proposed adaptive group normalization (AdaGN), respectively. Finally, we use a mutual information loss (info loss) to disentangle the intra-group and inter-group control [17].

## 3.1 Similarity-based Grouping Module

We group channels according to the similarity of convolutional kernels. Note that we use convolutional kernels instead of feature channels because the similarity calculated by feature channels would be imprecise in each training batch due to the limited batch size (e.g., 32). The convolutional kernels in a layer can be denoted as $\mathbf{W} \in \mathbb{R}^{c \times d}$, where $c$ and $d$ are the number of kernels and dimension of kernels, respectively. Since each kernel controls a channel in the output feature, our target is to cluster kernels with high similarity into a group. We propose to learn a variable $\mathbf{G} \in \mathbb{R}^{c \times g}$ to represent the group information, where $g$ indicates the number of groups. Each row of $\mathbf{G}$ is supposed to be a one-hot vector that indicates which group the corresponding channel belongs to. Such one-hot vectors can be obtained by an argmax function. As the argmax is not differentiable, we follow [30] to conduct a softmax function with a small temperature over each row of $\mathbf{G}$. The core idea of our grouping mechanism is that the similarity of the group information for two channels should consist with the similarity of the kernels for these channels. The similarity of two kernels can be obtained by:

$$S_{i,j} = \frac{\mathbf{w}_i^T \mathbf{w}_j}{\|\mathbf{w}_i\|_2 \cdot \|\mathbf{w}_j\|_2},$$ (1)

where $\mathbf{w}_i$ and $\mathbf{w}_j$ indicate the $i^{th}$ kernel and the $j^{th}$ kernel, respectively, and $S_{i,j}$ denotes corresponding cosine similarity. To fit the similarity matrix of $\mathbf{G}$ to the one of $\mathbf{W}$, we optimize $N(\mathbf{G}/t)N(\mathbf{G}/t)^T \rightarrow \mathbf{S}$, where $N$ is a softmax function and $t$ is a temperature.

Since we conduct softmax operations over each row of $\mathbf{G}$ to approximate one-hot vectors, the similarity matrix is also supposed to be a zero-one matrix. We realize this by conducting a unit step function with a threshold. Thus the supervision for $\mathbf{G}$ can be denoted as:

$$L_g = MSE(N(\mathbf{G}/t)N(\mathbf{G}/t)^T, H(\mathbf{S} - r)),$$ (2)

where $MSE(\cdot, \cdot)$ is a mean square error loss function, $N(\cdot)$ and $H(\cdot)$ are the softmax and unit step function, respectively, $t$ is a temperature, and $r$ is a threshold. The details for setting $t$ and $r$ can be found in Section 4. Such a grouping loss would not calculate gradient over $\mathbf{S}$ since it acts as a reference in this equation. Since each channel is supposed to belong to only one group, we conduct an argmax operation over each row of $\mathbf{G}$ to obtain grouping results.

## 3.2 Semantic-aware Embedding

In this subsection, we introduce details of semantic-aware embedding. The semantic-aware embedding makes use of semantic groups to embed latent codes into an intermediate latent space to further achieve semantic-aware control. Specifically, the proposed embedding consists of an intra-group embedding and an inter-group embedding. Inspired by StyleGAN [2], we first use a Multilayer Perceptron (MLP) to embed latent codes into $\mathbf{z} = (\mathbf{z}_s, \mathbf{z}_b)$ ($\mathbf{z}_s \in \mathbb{R}^c$ for standard deviation and $\mathbf{z}_b \in \mathbb{R}^c$ for mean) for inta-group control and $\mathbf{h} = (\mathbf{h}_s, \mathbf{h}_b)$ (where, $\mathbf{h}_s, \mathbf{h}_b \in \mathbb{R}^g$) for inter-group control. $\mathbf{z}$ and $\mathbf{h}$ are further embedded into generator based on semantic groups. We use $\mathbf{G} \in \mathbb{R}^{c \times g}$ to denote the grouping results of SGM for simplicity.

**Intra-group embedding**: We control intra-group channels with $\mathbf{z}$ via Group AdaIN, which conducts AdaIN for the style and the features in each group. Given a convolution feature $\mathbf{X}^{(n)}$ in the $n^{th}$ layer, we omit subscript $n$ for simplicity in the following introduction. Both $\mathbf{z} \in \mathbb{R}^c$ and $\mathbf{X} \in \mathbb{R}^{c \times HW}$ are corresponding to the channels grouped by SGM, thus the $i^{th}$ group of the latent code and the input feature are denoted as $\mathbf{z}_i = \mathbf{g}_i \circ \mathbf{z}$ and $\mathbf{X}_i = \mathbf{g}_i \otimes \mathbf{X}$, respectively where $\mathbf{g}_i \in \mathbb{R}^c$ indicates the $i^{th}$ column vector of $\mathbf{G}$, $\circ$ denotes a Hadamard product operation, $\otimes$ denotes a broadcast product

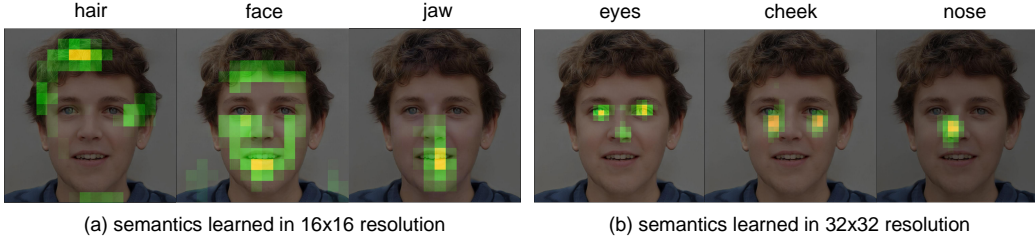

| hair | face | jaw | eyes | cheek | nose |

(a) semantics learned in 16x16 resolution    (b) semantics learned in 32x32 resolution

Figure 3: Visualization of the semantics learned in different resolutions. The attention maps are obtained by averaging the feature maps in a group. For clear representation, the tags are associated by human based on the responses to different region.

operation, e.g., $\mathbf{g}_i \otimes \mathbf{X} := diag(\mathbf{g}_i)\mathbf{X}$. We can use $\mathbf{z}_i$ to control the feature statistics of $\mathbf{X}_i \in \mathbb{R}^{c \times HW}$ via AdaIN, and the output feature is denoted as $\mathbf{X}'_i \in \mathbb{R}^{c \times HW}$:

$$\mathbf{X}'_i = \text{AdaIN}(\mathbf{z}_i, \mathbf{X}_i) = [z^i_{s,j} \frac{\mathbf{x}^i_j - \mu(\mathbf{x}^i_j)}{\sigma(\mathbf{x}^i_j)} + z^i_{b,j}], \forall j \in \{j : G_{j,i} = 1\}, \tag{3}$$

$\mathbf{x}^i_j \in \mathbb{R}^{HW}$ is the $j^{th}$ channel of $\mathbf{X}_i$, $z^i_{s,j}$ and $z^i_{b,j}$ are the $j^{th}$ element of the embedded latent code $\mathbf{z}_{i,s}$ and $\mathbf{z}_{i,b}$, respectively, $\mu(\cdot)$ and $\sigma(\cdot)$ calculate the mean and standard deviation for the input variable. In such a way, we can control the $i^{th}$ semantic by the embedded intra-group code $\mathbf{z}_i$.

**Inter-group embedding**: inter-group style $\mathbf{h} = (\mathbf{h}_s, \mathbf{h}_b)$ controls the relative importance of different semantic groups by our proposed Adaptive Group Normalization (AdaGN), which conducts normalization and modulation in a group level (AdaIN is in a channel level):

$$\mathbf{X}'' = \sum_{0 \leq i < g} \text{AdaGN}(\mathbf{h}_i, \mathbf{X}'_i) = \sum_{0 \leq i < g} h_{s,i} \frac{\mathbf{X}'_i - \mu(\mathbf{X}'_i)}{\sigma(\mathbf{X}'_i)} + h_{b,i}, \tag{4}$$

where $\mathbf{X}'_i \in \mathbb{R}^{c \times HW}$ is obtained in Equation 3, and $h_{s,i}$ and $h_{b,i}$ are the $i^{th}$ element of the embedded latent code $\mathbf{h}_s$ and $\mathbf{h}_b$, respectively. In such an operation, we remove the feature statistics in a semantic-aware way by group normalization and control inter-group relative importance by the latent code $\mathbf{h}$, which enables us to achieve fine-grained semantics control.

An extension of styleGAN claimed that conducting instance normalization over features would cause artifacts in generative models, and the normalization should be implemented on weights [12]. Our semantic-aware embedding can be equivalently converted to a weight normalization version by formulating a combination of $\mathbf{z}$ and $\mathbf{h}$:

$$\mathbf{f} = \sum_{0 \leq i < g} \frac{h_i(\mathbf{z} \circ \mathbf{g}_i)}{\sigma((\mathbf{z} \circ \mathbf{g}_i) \otimes \mathbf{X})} = \sum_{0 \leq i < g} \frac{h_i(\mathbf{z} \circ \mathbf{g}_i)}{\sqrt{\frac{1}{c} \sum_{0 \leq j < c} (z_j G_{j,i})^2}}, \tag{5}$$

where $\circ$ denotes a Hadamard product operation, $\otimes$ denotes a broadcast product operation, $\mathbf{g}_i$ indicates the $i^{th}$ column vector of $\mathbf{G}$, $G_{j,i}$ indicates the element in the $j^{th}$ row and $i^{th}$ column, and $\mathbf{f} \in \mathbb{R}^c$ is a combination of $\mathbf{z}$ and $\mathbf{h}$. Note that the last item is independent from $\mathbf{X}$, because the row vectors of $\mathbf{X}$ are assumed to be i.i.d. random variables with unit standard deviation [12]. Once we get $\mathbf{f}$, we can simply follow [12] to obtain the weight normalization version:

$$w'_{ijk} = f_i \cdot w_{ijk}, \quad w''_{ijk} = \frac{w'_{ijk}}{\sqrt{\sum_{i,k} w'_{ijk}{}^2 + \epsilon}}, \tag{6}$$

where $f_i$ is the $i^{th}$ element of $\mathbf{f}$, $w_{ijk}$ denotes convolution weights, $w'_{ijk}$ denotes modulated weights, $w''_{ijk}$ denotes demodulated weights, and $\epsilon$ is a small constant to avoid numerical issues.

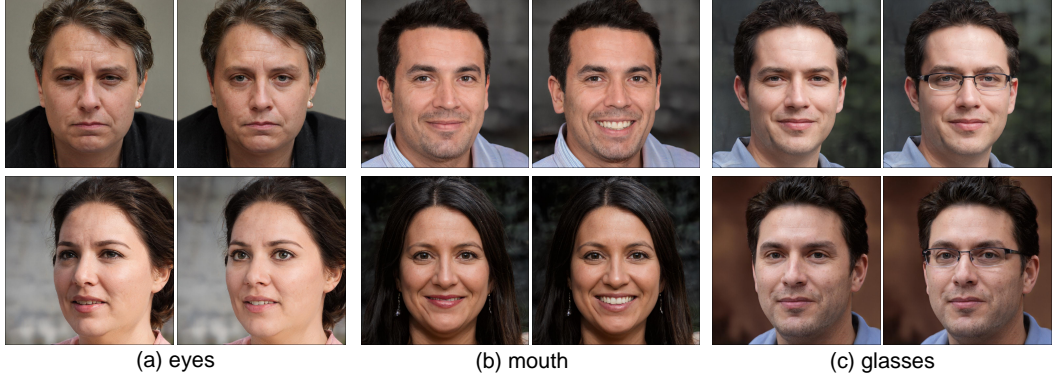

|       (a) eyes       |       (b) mouth       |       (c) glasses       |

Figure 4: An illustration of semantic-specific manipulation based on intra-group codes. We can control the appearance of fine-grained semantics by adjusting intra-group relative importance. Specifically, we visualize the image attributes controlled by each semantic group and change the corresponding intra-group latent codes. The tags (i.e., eyes, mouths, and glasses) are associated by human.

### 3.3 Objective Functions and Optimization

The proposed Similarity-based Grouping Module and Semantic-aware Embedding Module enable the SariGAN to achieve semantic-aware control via embedding latent codes into a semantic-aware intermediate latent space ($[\mathbf{z}, \mathbf{h}]$). To further disentangle the factors in such a space, we optimize the model with a mutual information loss, which can disentangle latent factors by maximizing the mutual information between the latent variables and observations [17].

Our full model is jointly optimized by adversarial training, a channel grouping loss, and a mutual information loss, which can be denoted as:

$$\min_{\mathbf{g}, M, G, Q} \max_{D} L(\mathbf{g}, M, G, Q, D) = L_a(D, G) + \lambda_1 L_g(\mathbf{g}) - \lambda_2 L_I(G, Q), \quad (7)$$

where $D, G, Q, \mathbf{g}, M$ are the discriminator, generator, auxiliary distribution, grouping parameters, and mapping networks respectively, $L_a(D, G)$ is the adversarial loss, $L_g(\mathbf{g})$ indicates the grouping loss in Equation 2, and the $L_I(G, Q)$ is a variational lower bound of the mutual information:

$$L_I(G, Q) = E_{[z,h] \sim P([z,h]), x \sim G([z,h])}[\log Q([z,h]|x)] + H([z,h]), \quad (8)$$

where $H(\cdot)$ is an entropy item $[z, h]$ are samples in the semantic-aware intermediate latent space.

## 4 Experiments

### 4.1 Experiment Setup

**Dataset** We conduct experiments on both unconditional image generation and conditional image inpainting. For unconditional image generation, we evaluate our SariGAN on three datasets, including LSUN CATS [31], LSUN CARS [31], and FFHQ [2] dataset. For image inpainting, we use Paris Street View [32] for evaluation. The details of these datasets can be found in Table 1.

| Task | Dataset | #Images | Resolution |
|---|---|---|---|
| Unconditional generation | LSUN_CATS [31] | 1.6M | $256 \times 256$ |
|  | LSUN_CARS [31] | 0.9M | $512 \times 384$ |
|  | FFHQ [2] | 70K | $1024 \times 1024$ |
| Inpainting | PSV [32] | 14.9K | $256 \times 256$ |

Table 1: Detailed statistics of the datasets used in this paper.

**Evaluation** For unconditional image generation, we adopt Frechet inception distance (FID) as our quantitative evaluation matrices, since FID has shown consistent performance with human perception [33]. To conduct qualitative analysis, we visualize randomly sampled images in Figure 6. We show results for semantic-specific control based on intra-group and inter-group codes in Figure 4 and

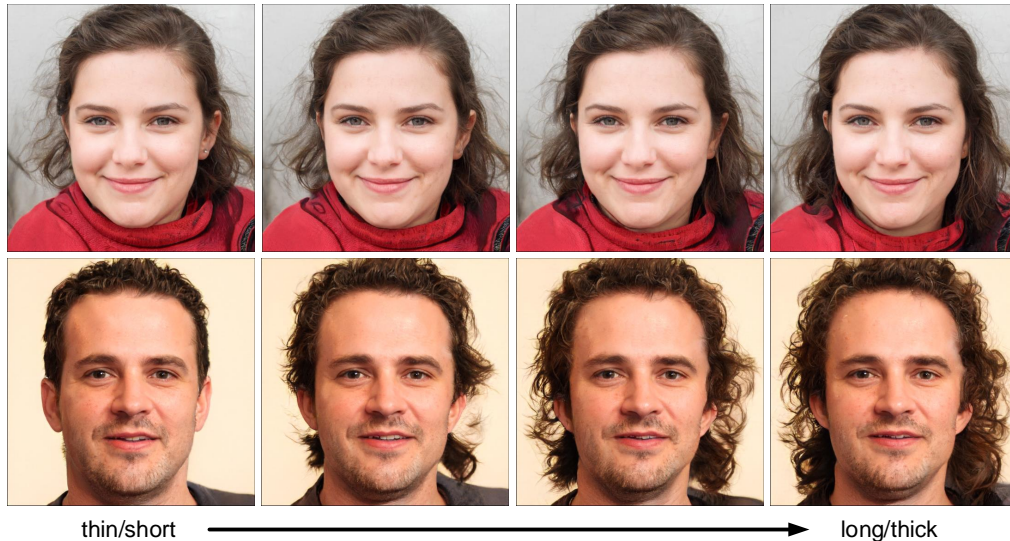

thin/short                                                 long/thick

Figure 5: An illustration of semantic-specific manipulation based on the inter-group codes. We can control the proportion of fine-grained semantics (e.g., hair and other facial attributes) by adjusting inter-group relative importance.

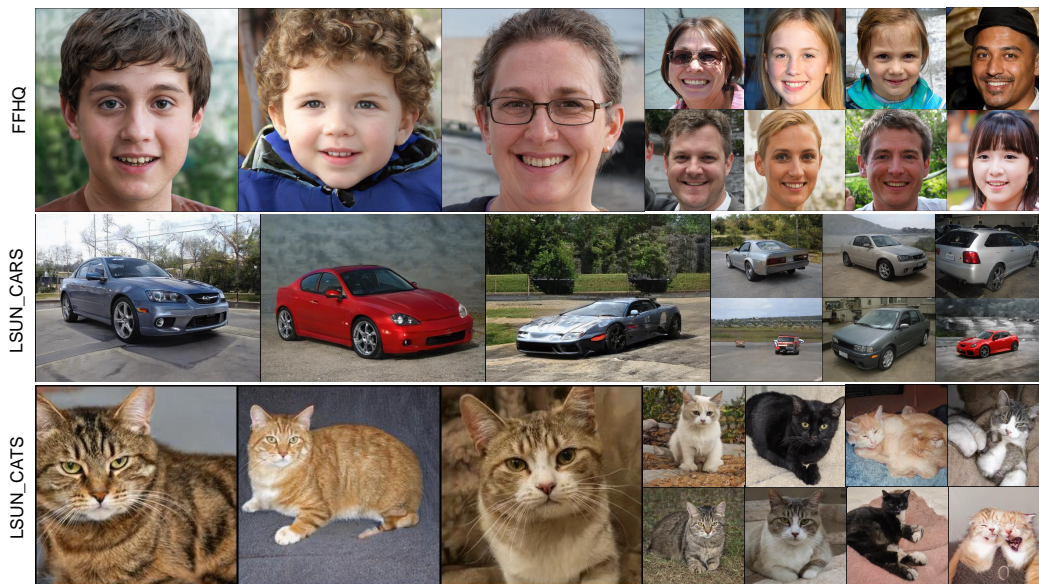

Figure 6: Uncurated results for LSUN CATS [31], LSUN CARS [31], and FFHQ [2].

Figure 5 respectively to visualize the disentanglement of fine-grained semantics. For image inpainting, we use mean-average-error (MAE) and FID to evaluate both per-pixel reconstruction accuracy and perceptual image quality of inpainting results.

**Implementation details** For data pre-processing, we follow StyleGAN [2] to pad training images in LSUN CARS [31] from $512 \times 384$ to $512 \times 512$ with zeros. The choice of hyperparameters: a smaller $t$ in Equation 2 would better approximate one-hot vector, while also makes the optimizing harder since the gradient would vanish. We defer such a trade-off to the network by making $t$ learnable, and the learned $t$ is around 0.3. The setting of the threshold $r$ (e.g.,0.1) in Equation 2 is conditioned on the group number $g$ (e.g.,16), and the principle is that each channel would have $c/g$ similar channels after activated on average. The $\lambda_1$ and $\lambda_2$ in Equation 7 is experimentally set to be 2 and 10, respectively. For the implementation of info loss, we follow [17] to make $Q$ and $D$ share all convolutional layers.

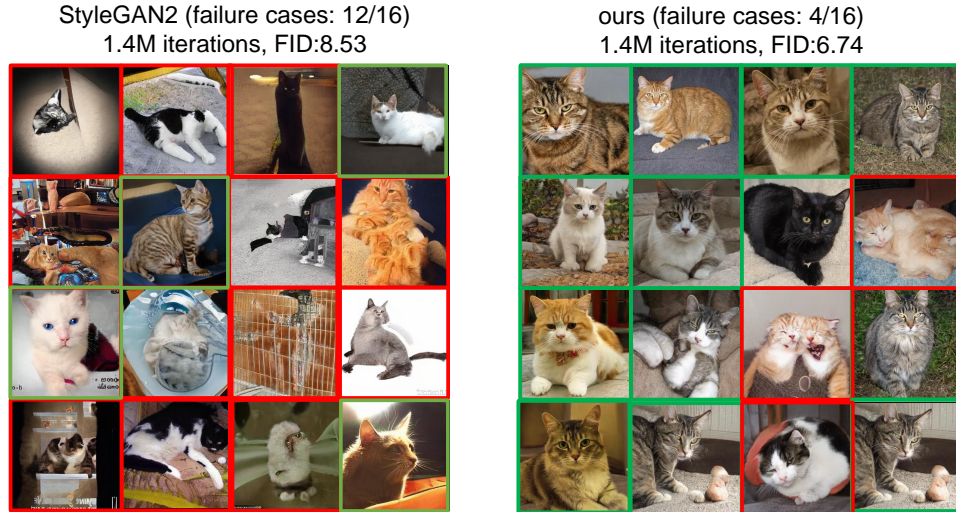

StyleGAN2 (failure cases: 12/16)
1.4M iterations, FID:8.53

ours (failure cases: 4/16)
1.4M iterations, FID:6.74

Figure 7: We compare our model with the StyleGAN2 on the LSUN CATS dataset. Our model can significantly outperform the StyleGAN2 on generating natural and realistic images.

Similar to [2], the mapping network in SariGAN consists of eight layers and the synthesis network consists of two layers for each resolution ($4^2256^2/512^2/1024^2$). The semantic-aware embedding is conduced in each layer of the synthesis network, except for the first layer and to-RGB layers, and a detailed discussion can be found in supplementary material. The discriminator consists of 16, 18, and 20 layers for the CATS, CARS, and FFHQ datasets, respectively (i.e., two layers for each resolution $4^2 - 256^2/512^2/1024^2$ and two additional layers). We use PyTorch [34] as our codebase and run each experiment on 8 Tesla V100 GPUs for 7 days. More details can be found in our code `https://github.com/researchmm/SariGAN`.

| Model | FID |
|---|---|
| Baseline (StyleGAN2 [2]) | 8.16 |
| Ours w/o Info loss | 7.77 |
| Ours w/ Info loss | **6.74** |

Table 2: Ablation study for info loss on LSUN CATS [31] in terms of FID.

| Model | FID |
|---|---|
| Baseline (StyleGAN2 [2]) | 8.16 |
| Ours w/ 8 groups | 7.62 |
| Ours w/ 16 groups | **6.74** |
| Ours w/ 32 groups | 7.74 |

Table 3: Ablation study for group number on LSUN CATS [31] in terms of FID.

| Model | CATS | CARS |
|---|---|---|
| StyleGAN [2] | 8.53 | 3.27 |
| StyleGAN2[12] | 8.16 | 3.01 |
| Ours | **6.74** | **2.98** |

Table 4: Quantitative comparisons in terms of FID. For fair comparison, we implement StyleGAN2[12] config-E with the same codebase and compare the results in the same training iteration (i.e., 1.4M iterations for CATS and 1M iterations for CARS).

| Method | FID | MAE |
|---|---|---|
| CA[35] | 30.38 | 0.0338 |
| AN[36] | 44.69 | 0.0334 |
| StyleGAN2 [12] | 36.12 | 0.0307 |
| Ours | **29.49** | **0.0298** |

Table 5: Quantitative comparisons on PSV [32]. Our model outperforms baselines in terms of per-pixel reconstruction accuracy (MAE) and perceptual image quality (FID).

## 4.2 Unconditional image generation

Unconditional image generation (synthesis) is the task of generating new images unconditionally from an existing dataset. In this subsection we discuss a set of experiments on unconditional image generation to evaluate the quantitative and qualitative quality of our results.

**Quantitative evaluation**: Table 2 shows the ablation study on intra/inter-group embedding and info loss. It can be observed that the proposed intra/inter-group embedding can improve the performance by 0.4 FID, and the info loss can further obtain 1.0 improvements by disentangling fine-grained semantics. The number of groups is an important hyper parameter in our model, thus we conduct

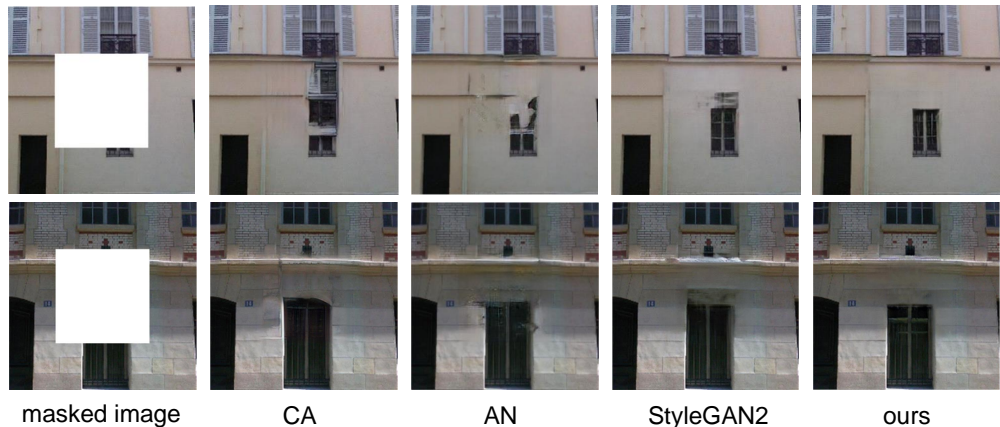

| masked image | CA | AN | StyleGAN2 | ours |

Figure 8: Qualitative comparison on PSV [32] with the SOTA inpainting models CA [35], AN [36], and styleGAN [12]. Our model is able to generate plausible structures and fine textures for the tree in results. More cases can be found in supplementary material. [Best viewed with zoom-in]

ablation study in Table 3, which shows 16 groups can achieve the best performances. We compare our proposed model with the SOTA StyleGAN [2] and StyleGAN2 [12] in Table 4, and our models can outperforms StyleGAN2 [12] with a clear margin especially on CATS dataset, which is more challenging than CARS dataset. Moreover, we compare our semantic grouping module with the hand-crafted block diagonal constraint proposed in DBT [15], we observed 11.9% relative improvements in terms of FID on LSUN CATS by our SGM. More ablations and comparisons on the three datasets can be found in supplementary material.

**Qualitative evaluation**: First, we visualize the learned semantics in different resolutions in Figure 3, which are quite intuitive. More visualizations are discussed in supplementary material. To visualize the intra-/inter-group control, we conduct style mixing [2] by replacing the $z_i/h_i$ of an image to that of another image with different attributes. The results in Figure 4 and Figure 5 show that $z$ can influence the appearance of a semantic (e.g., the color and shape of eyes) and $h$ can influence the proportion of a semantic (e.g., the length of hair). Moreover, we randomly select the generated images to show the generation quality, which can be found in Figure 6 and Figure 7.

### 4.3 Conditional image inpainting

Image inpainting is a task that takes masked images and corresponding masks as conditions to complete missing regions in the damaged input images [37]. We report quantitative comparisons in Table 5 and qualitative results in Figure 8. Specifically, we extend StyleGAN and SariGAN by adding an encoder to encode the conditions for generating missing contents (more details see supplementary material). The table shows that our model outperforms baselines in both per-pixel reconstruction accuracy (MAE) and perceptual image quality (FID). And Figure 8 shows that SariGAN is able to generate plausible structures and fine textures in image inpainting.

## 5 Conclusion

In this paper, we propose to learn semantic-aware relative importance in GAN (SariGAN) for fine-grained semantic disentanglement. In comparison to previous style-based generators, SariGAN is able to achieve finer-grained style control and higher fidelity of image synthesis without additional supervision. Experiments show that SariGAN achieves SOTA performance in both unconditional image generation and conditional image inpainting tasks. Considering that different semantics are separated by channel clustering, the channel clustering algorithm plays an important role in SariGAN. We consider two aspects to further improve SariGAN in future work. First, we will design to learn the number of groups in each layer instead of a handcrafted one. Second, the clustering results in high-resolution layers are not promising in current SariGAN. We will study further improvements on semantic clustering in GAN, which can improve the understanding and controllability of GAN.

## Acknowledgement

This work was supported by the National Key RD Program of China under Grant 2017YFB1300201, the National Natural Science Foundation of China (NSFC) under Grants U19B2038 and 61620106009, the Key Scientific Technological Innovation Research Project by Ministry of Education of China and the University Synergy Innovation Program of Anhui Province under GXXT-2019-025.

## Broader Impact

The proposed image generation approach pursues generating high-fidelity and various image samples. Extensive experiments have shown competitive performance on several benchmark datasets. This work has the potential positive impact of enabling automatic content creation and editing for high-quality multimedia data like images and videos, and generating image variants from existing training datasets at no additional cost to human annotators. By using the synthesized data, the proposed approach may also benefit the training of more computer vision tasks like image classification and object detection for better modeling large data variations in the real world.

At the same time, any image generation application runs the risk of producing biased or offensive content as the priors existed in human-curated training datasets. Besides, the rapid progress in image generation and manipulation has now come to a point where it raises significant concerns in our society. Leveraging powerful image generation technologies (including the proposed approach in this paper) to manipulate or generate visual content with a high potential to deceive might cause harm, especially by spreading fake information. More work is needed to build towards a more automated pipeline for image generation, review, and publishing.

## Footnotes

*This work was performed when Heliang Zheng and Yanhong Zeng were visiting Microsoft Research as research interns.

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
