[Supplementary Material]

# Supplementary Material: Learning Semantic-aware Normalization for Generative Adversarial Networks

## 1  Unconditional Image Generation

**Ablation study**: We compare our proposed Similarity-based Grouping Module (SGM) with random grouping on LSUN CATS [26] dataset in Table 1 (140w iterations with batch size of 32). It can be observed that random grouping obtains improvements compared to StyleGAN2 [7], which benefits from the mechanism of intra-/inter group embedding and info loss. The proposed SGM can further improve the performance with an obvious margin, which shows the importance of learning semantics. Table 2 shows the results of conducting semantic-aware control at different resolutions (40w iterations). We use FFHQ [2] dataset with $1024 \times 1024$ image resolution, so that a large range of settings can be studied. Features with low resolutions (e.g., $8 \times 8 - 64 \times 64$) show better performance on learning semantics (refer to Figure 1), and $256 \times 256$ is the most common resolution used in image generation. Thus we conduct experiments with three settings: $8 \times 8 - 64 \times 64$, $8 \times 8 - 256 \times 256$, and $8 \times 8 - 1024 \times 1024$. It can be observed that semantic-aware control for low-resolution features can improve the performance. While for high-resolution features whose semantics are hard to learn, semantic-aware control would cause a performance drop.

**Visualization**: We show attention maps of the learned semantics at different resolutions (from $8 \times 8$ to $256 \times 256$) in Figure 1. It can be observed that features with low resolutions (e.g., $8 \times 8 - 64 \times 64$) show better performance on learning semantics. Figure 2 shows the semantic interpolation results. Specifically, we visualize the image attributes controlled by each semantic group, and obtain Figure 2 by conducting interpolation in the latent space of the corresponding group. It can be observed that we can realize independent control on fine-grained semantics by the proposed SariGAN. Figure 3 shows the qualitative comparison of scale-specific control by StyleGAN2 and semantic-specific control by SariGAN. The results are obtained by style-mixing, which replaces the corresponding latent codes of the source images with that of the reference images. It can be observed that SariGAN can control a specific semantic (e.g., mouth) while preserving identities.

## 2  Conditional Image Inpainting

Image inpainting is a task that takes both masks and masked images as conditions to complete missing regions in input images. To extend SariGAN for image inpainting, we add an image encoder to encode masks and masked images. The encoded features are then used as the input of the generator and the mapping network in the extended model. Style codes from the mapping network are used to control features in each layer of the generator. Besides the adversarial loss, the channel grouping loss, and the mutual information loss mentioned in Section 3.3 in the paper, we also use L1 loss as a reconstruction loss, which is widely used in image inpainting [27,30,31].

We evaluate models by using central square masks on Paris Street View [27] following the common setting used in most inpainting papers [27,30,31]. All images are cropped and resized to $256 \times 256$ for both training and testing. We show more qualitative comparison results with SOTA inpainting models in Figure 4 [27,30,31]. Through specially-designed network architecture and optimization, SariGAN is able to achieve finer-grained style control and higher fidelity of unconditional image synthesis, which also brings benefits in image inpainting. Visual results in Figure 4 show that our model achieves SOTA performance in image inpainting.

| Model | FID |
|---|---|
| Baseline (StyleGAN2 [7]) | 8.16 |
| Ours w/ random grouping | 7.35 |
| Ours w/ semantic grouping (SGM) | **6.74** |

Table 1: Comparison of baseline, random grouping and semantic grouping (i.e., the proposed SGM) on LSUN CATS [26] in terms of FID.

| Model | FID |
|---|---|
| Baseline (StyleGAN2 [7]) | 3.95 |
| Ours $8 \times 8 - 64 \times 64$ | **3.92** |
| Ours $8 \times 8 - 256 \times 256$ | 4.26 |
| Ours $8 \times 8 - 1024 \times 1024$ | 4.85 |

Table 2: Conduct semantic-aware control at different resolutions on FFHQ [2] in terms of FID.

Figure 1: Visualization of the semantics learned in different resolutions. We show 16 groups in each layer with the resolution increasing from $8 \times 8$ to $256 \times 256$. The attention maps are obtained by averaging the feature maps in a group. It can be observed that features with low resolutions (i.e., $8 \times 8 - 64 \times 64$) show better performance in learning semantics (e.g., eyes, mouths and hair).

Glasses

Mouth

Eyes

Beard

Figure 2: An illustration of semantic-aware disentanglement. We can realize independent control on fine-grained semantics by conducting interpolation in latent space. Specifically, we visualize the image attributes controlled by each semantic group, and conduct interpolation in the latent space of the corresponding group. For clear representation, the tags (i.e., glasses eyes, mouths, and beard) are associated by human.

| Source | Reference | StyleGAN2 | Ours |
|---|---|---|---|

Figure 3: Qualitative comparison of StyleGAN2 and SariGAN by controlling the semantic of mouth. SariGAN (semantic-specific control) can control a specific semantic while preserving identities. StyleGAN2 (scale-specific control) makes several unexpected changes, as the semantics of mouth, eyes, and hairstyle are highly-entangled.

| Groud truth | Input | CA | AN | StyleGAN | Ours |

Figure 4: Qualitative comparisons on Paris Street View [27] with SOTA inpainting models CA [30], AN [31], and StyleGAN [7]. Results show that our model is able to generate plausible structures of windows and fine-grained details of trees and achieve SOTA performance in image inpainting.