[Reviews · NeurIPS 2020]

Review 1

Summary and Contributions: This paper improves the StyleGAN-based image generation model by disentangling semantics based on a learnable semantics grouping operation, where the styles of the intra-group features are controlled by group-wise adaptive instance normalization and the overall features are re-balanced by inter-group adaptive group normalization. Quantitative and qualitative evaluations show certain improvements over existing methods.

Strengths: - The quantitative evaluations and ablation study validates the effectiveness of the proposed improvements.

Weaknesses: 1. The most critical limitation of this work is its novelty and theoretical soundness. - Disentanglement by grouping similar features is not new and extensive discussed in existing works, such as DBT [10]. However, similarity between layers of a convolutional kernel may not indicate consistent similarity between corresponding feature channels. If not so, the authors should prepare more evidences or discussions. - Mutual information loss, or the framework of InfoGAN, is well-explored in controllable GAN methods. Just employing it to regularize style codes may not be qualified as a valid contribution. [After rebuttal] The feedback discussed the differences with DBT: 1) employing different grouping strategies (uniformly divided channels v.s. adaptively clusterred channels) and 2) being designed for different tasks (classification v.s. unsupervised image generation). I am happy that an additional baseline (DBT as the grouping algorithm) was reported, with measurable performance drop against the proposed method. Thus although the channel-grouping may not new, the proposed group strategy together with intra- and inter-group association is useful in this specific task. 2. Another weakness lies on the qualitative comparison. The functionalities of the proposed modules are hard to observe from these comparisons. For example, - Besides numerous examples to tell the semantics disentanglement of the proposed method, such as Fig. 4 and Fig. 5, there lacks necessary visual comparisons with previous methods. In Fig. 1, the differences between StyleGAN v2 and the proposed method is too subtle as well. - Even the quantitative metrics shown better scores, the inpainting result in Fig. 6 does not have appealing improvement than previous methods. [After rebuttal] I am happy that the author feedback provided more visual results, which are illustrative and helpful.

Correctness: The claims and method may be correct but not convincing right now. They need more in-depth discussions and evaluations.

Clarity: This paper is generally organized well. But there contain numerous typos and grammar errors in the paper. For example, ``hline'' in table 3. ``40w iteration'' -> ``400k iterations'', and etc.

Relation to Prior Work: As indicated above, more discussions about the difference between the previous works are expected in the related work.

Reproducibility: Yes

Additional Feedback:


Review 2

Summary and Contributions: This paper proposes a similarity-based grouping method for semantic disentanglement in GAN. The core idea lies in clustering channels with the same semantics according to similarity and adaptive group-wise normalization. Experiments are conducted on four datasets and achieved superior performance compared to other methods. In addition, ablation studies are conducted to verify the effectiveness of the proposed method.

Strengths: The method part solved the problem of fine-grained semantic disentanglement based on semantic-aware relative importance. The proposed intra-group and intro-group embedding make use of intermediate latent space to achieve semantic-aware control. Evaluation on LSUN CATS, LSUN CARS, FFHQ, and Paris Street View datasets demonstrate the effectiveness of the proposed method.

Weaknesses: 1. There should be a formal definition of the task in the paper, i.e., unconditional image generation and inpainting. 2. The experimental part does not cover some hyper-parameters in the proposed similarity-based grouping method such as \lambda_1 and \lambda_2 in equation (7).

Correctness: The claims and method are correct.

Clarity: The paper is clearly presented. It would be better to further smooth the the relationship between section 3.1, 3.2, and the objective functions in section 3.3.

Relation to Prior Work: The authors introduce style-based generator and interpretable representation in the related work part. I think the introduction to the prior work is sufficient.

Reproducibility: Yes

Additional Feedback: It is expected to compare the contribution of intra-group and intro-group embedding to the final result, so as to show the effectiveness of each module in the SariGAN.


Review 3

Summary and Contributions: This paper focuses on learning to disentangle latent factors in image generation tasks. Considering that existing work “styleGAN” can only embed latent code into different image resolutions and control scale-aware image styles, the authors propose to learn a semantic-aware manipulation (on feature channels) by a learned AdaGN operation. Particularly, this disentanglement is achieved by proposing a feature channel clustering module and embedding latent codes into both intra-groups and inter-groups. The resultant synthesized images show SOTA performance on a broad range of image synthesis tasks, including unconditional image generation (on CATs, CARs, and FFHQ face), and conditional image inpainting tasks.

Strengths: 1. This paper proposes to solve a key problem that the latent space in image generation can be further disentangled (not only scale-aware, but semantic-aware) for high-quality generation results and good interpretable properties. 2. The framework of designing channel similarity grouping, and sematic-aware mapping (into both intra- and inter-group) are novel. I think this is a reasonable and solid design for learning fine-grained semantics. 3. The paper is well-written, and the experimental comparison, ablation study, and case studies are comprehensive in validating the effectiveness of the proposed method.

Weaknesses: 1. This paper is suggested to add some details on the discriminator architectures (e.g., how many layers used for different datasets) 2. Will the code be released in the future for a better reproducibility?

Correctness: Yes. The claims and the method are correct. The empirical methodology is correct.

Clarity: Yes. this paper is very clear and well-written.

Relation to Prior Work: Yes. The differences between this work and some previous contributions are clearly discussed in this paper.

Reproducibility: Yes

Additional Feedback: [After rebuttal] After reading the rebuttal and other reviewers' comments, I keep my original rate.

[Author Response · NeurIPS 2020]

We sincerely thank all reviewers and appreciate the positive comments on "a solid design", "solving a key problem"
and "comprehensive experiments". In the following, we address the concerns from each reviewer.

**To Reviewer #1:**

**Q 1.1 Novelty.** There are two key differences between DBT [10] and our work. 1) Different tasks. Our SariGAN aims
to disentangle fine-grained semantics for unsupervised generative models. We achieve this by mining (with SGM) and
advancing (with AdaGN) the intrinsic attributes of channels (i.e., semantics) in the latent space of relative importance
(line 50-51 in the paper). While DBT [10] proposed to learn group bilinear features for classification. 2) Different
grouping algorithms. DBT [10] is not designed for generation tasks. For example, the uniformly divided channels (per
group) prohibit learning from more channels to represent complex semantics. Also, the hand-crafted block diagonal
constraint propagates inconsistent gradient against the generation task. We have observed an 11.9% relative drop in
terms of FID on LSUN CATS if using DBT [10] as grouping algorithms. We will add this discussion to related work.

For InfoGAN loss, it is a general framework designed for latent factors disentangling. We use InfoGAN in the paper for
ensuring that inter/intra-group semantics can be well-disentangled. In Table 2, experiments demonstrate the gain of
16% achieved by using infoGAN loss. We believe such improvement is non-trival, and the using of InfoGAN is worth
mentioning. We will take your helpful suggestions and make the statement more clear in the final version.

**Q 1.2 The relation of kernels' similarity and feature channels' similarity.** Given a well-trained model, the semantic
of each feature channel is decided by the corresponding kernel. In experiments, we randomly generate 10k samples to
calculate the pairwise similarity of feature channels and compare it with the similarity of kernels. As shown in Figure 1
(a), kernels' similarity and feature channels' similarity are positively correlated. Note that if we directly use feature
channels, the similarity would be imprecise in each training batch due to the limited batch size (e.g., 32). Specifically,
according to the law of large numbers, the more samples calculated, the more precise similarity approximated. As shown
in Figure 1 (b), the similarity of channels calculated in a batch causes severe inconsistency. We will add discussions on
this problem according to your kind suggestions.

Figure 1: An illustration of the similarity consistency
of kernels, feature channels, and feature channels in a
batch. The more diagonally concentrated, the better con-
sistency. It can be observed that the similarity of kernels
in (a) is more consistent with feature channels than that
of channels calculated in a batch in (b).

Figure 2: Qualitative comparison of StyleGAN2 and
SariGAN. SariGAN can control a specific semantic
(e.g., mouth) while preserving identities. StyleGAN2
makes several unexpected changes, as the semantics
of mouth, eyes, and hairstyle are highly-entangled.

**Q 1.3 Qualitative comparison.** We provide qualitative comparisons with StyleGAN2 in Figure 2. It can be observed
that SariGAN can achieve semantic-level controls (e.g., control the mouth), while StyleGAN2 can only achieve scale-
level controls (e.g., the semantics of mouth, eyes, and hairstyle are still entangled). More cases for semantic-specific
controls can be found in Figure 2 in our supplementary. Thanks for your valuable comments, and we will add more
qualitative comparisons in the final version.

For inpainting results, we provide more cases in Figure 4 in the supplementary material to eliminate case biases, which
show consistent qualitative improvements by SariGAN comparing over SOTA.

**To Reviewer #3:** Definition: Unconditional image generation (synthesis) is the task of generating new images
unconditionally from an existing dataset. And image inpainting aims at filling missing pixels in a damaged image given
a corresponding mask (line 220). The $\lambda_1$ and $\lambda_2$ in Equation 7 are set to be 2 and 10, respectively (line 191).

The inter-group and intra-group embeddings disentangle semantics in different levels. The former controls semantics
like pose, age, and gender (Figure 4 in the paper); and the latter controls semantics like mouth, eyes, and glasses (Figure
2 in our supplementary). Thanks for your valuable suggestions, and we will make these clearer in the final version.

**To Reviewer #4:** Thanks for your valuable comments. The discriminator consists of 16, 18, and 20 layers for the
CATS, CARS, and FFHQ datasets, respectively (i.e., two layers for each resolution $4^2 - 256^2/512^2/1024^2$ and two
additional layers). We will add all these details in the final version and release both codes and models.

[Meta-Review · NeurIPS 2020]

R3 and R4 rate the paper top 50% papers, while R1 votes the paper marginally below the bar. While R1 initially raised several concerns on the paper's novelty side, R1 upgrades the rating of the paper since the rebuttal addresses the concerns. After consolidating the reviews and rebuttal, the AC finds the proposed method interesting. The channel grouping and normalization based on the filter similarity is new for generator design, and the results and analysis presented in the paper support the claim. The AC determines that the paper has merits to be published in the NeurIPS conference and would like to recommend its acceptance.